# Kismet/CHD7/CHD8 and Amyloid Precursor Protein-like Regulate Synaptic Levels of Rab11 at the *Drosophila* Neuromuscular Junction

**DOI:** 10.3390/ijms25158429

**Published:** 2024-08-01

**Authors:** Emily L. Hendricks, Nicole Linskey, Ireland R. Smith, Faith L. W. Liebl

**Affiliations:** Department of Biological Sciences, Southern Illinois University Edwardsville, Edwardsville, IL 62026, USA

**Keywords:** synapse, *Drosophila* neuromuscular junction, amyloid precursor protein, Rab11, Kismet, CHD7, CHD8, Alzheimer’s disease, chromatin remodeling

## Abstract

The transmembrane protein β-amyloid precursor protein (APP) is central to the pathophysiology of Alzheimer’s disease (AD). The β-amyloid hypothesis posits that aberrant processing of APP forms neurotoxic β-amyloid aggregates, which lead to the cognitive impairments observed in AD. Although numerous additional factors contribute to AD, there is a need to better understand the synaptic function of APP. We have found that *Drosophila* APP-like (APPL) has both shared and non-shared roles at the synapse with Kismet (Kis), a chromatin helicase binding domain (CHD) protein. Kis is the homolog of CHD7 and CHD8, both of which are implicated in neurodevelopmental disorders including CHARGE Syndrome and autism spectrum disorders, respectively. Loss of function mutations in *kis* and animals expressing human APP and BACE in their central nervous system show reductions in the glutamate receptor subunit, GluRIIC, the GTPase Rab11, and the bone morphogenetic protein (BMP), pMad, at the *Drosophila* larval neuromuscular junction (NMJ). Similarly, processes like endocytosis, larval locomotion, and neurotransmission are deficient in these animals. Our pharmacological and epistasis experiments indicate that there is a functional relationship between Kis and APPL, but Kis does not regulate *appl* expression at the larval NMJ. Instead, Kis likely influences the synaptic localization of APPL, possibly by promoting *rab11* transcription. These data identify a potential mechanistic connection between chromatin remodeling proteins and aberrant synaptic function in AD.

## 1. Introduction

Alzheimer’s disease (AD) is characterized by the loss of synapses and neurons in the cortex and hippocampus leading to cognitive impairments [1]. Neuronal dysfunction in AD is correlated with the accumulation of synaptic amyloid beta (Aβ) oligomers [2], which are generated by cleavage of the transmembrane protein, amyloid precursor protein (APP) [3]. APP is composed of a large extracellular domain, which may function in cell adhesion, ligand binding, and dimerization, a transmembrane domain, and a short C-terminus [3]. Cleavage by α- or β-secretase produces the soluble ectodomains, APPα or APPβ, respectively. Subsequent cleavage of αAPP by γ-secretase releases the P3 peptide in the non-amyloidogenic pathway while βAPP cleavage by γ-secretase yields Aβ in the amyloidogenic pathway. Both pathways produce an APP intracellular domain (AICD), which has putative roles in intracellular signaling [4]. Both the synaptic levels of APP [5] and the localization of APP to endosomes versus plasma membrane [6,7] influence Aβ production.

APP, along with the related proteins APP-like (APPL) 1 and APPL2, play partly functionally redundant roles in cell adhesion, proliferation, differentiation, neurite outgrowth, and synaptogenesis [3]. In the developing nervous system, the initial expression of APP and APPL correlates with neurogenesis [8], neurite outgrowth, and synaptogenesis [9]. APP is expressed in both pre- and postsynaptic neurons and loss-of-function mutations reduce hippocampal CA1 and cortical dendritic complexity in older animals [10]. Similarly, fly *appl^d^* loss-of-function mutants exhibit a significant reduction in the number of boutons at the neuromuscular junction (NMJ) [11]. APP is important for synaptic plasticity and memory in mice [12,13] and flies [14] suggesting critical synaptic functions. Decades of research, however, have failed to yield a clear understanding of the non-pathological roles of APP. Further, focusing on APP and Aβ production has demonstrated little therapeutic benefit for AD patients [15]. AD is multi-faceted and includes perturbations in cytoskeletal dynamics, endolysosomal trafficking, lipid homeostasis, mitochondrial function, RNA splicing, microglial immune activation, and blood–brain barrier integrity [16].

To better understand the pathology of AD, next-generation sequencing and transcriptomic analyses have identified disrupted genomic loci in affected patients. Hundreds of mutations are correlated with late-onset AD including genes that regulate synaptic transmission (downregulated), biological adhesion (upregulated) [17,18,19], and the synaptic vesicle pathway [20]. The altered transcriptional programs associated with AD are caused by structural modifications to chromatin, which were first documented four decades ago [21,22]. Chromatin structure is established and maintained by a combination of epigenetic regulators, including chromatin modifying and chromatin remodeling enzymes, that collectively determine the accessibility of RNA polymerase II to proximal promoter sequences [23]. Expression of genes encoding enzymes that influence chromatin assembly and disassembly and nucleosome organization is altered in AD mouse models [24]. As such, epigenetic changes including DNA methylation and histone acetylation and methylation are found in AD patients and mouse models of AD [25]. 

Chromatin remodeling enzymes orchestrate epigenetic modifications by using ATP to slide nucleosomes relative to DNA, interchange histone protein subunits, and remove histones from DNA [23]. The chromodomain helicase DNA (CHD) binding protein family of chromatin remodelers plays critical roles in neurodevelopment and adult neurogenesis [26]. Although CHD proteins are better known for their roles in neurodevelopment including neural migration and differentiation [27,28], much less is known about their functions in mature neurons where they continue to be expressed [29,30]. 

CHD proteins and APP possess overlapping neuronal functions. Both CHD7 [31] and APP [32] are expressed in neural crest cells and are important for neural migration [31,33]. AD patients exhibit hyperacetylation of histone 3 lysine residue 27 (H3K27) loci in the entorhinal cortex that extensively overlap with loci affected in patients with Kallman Syndrome [17,34], a neurodevelopmental disorder correlated with *Chd7* mutations [35]. The link between AD phenotypes and aberrant histone acetylation has led to the use of histone deacetylase (HDAC) inhibitors to improve cognitive function [36]. Class I and II HDAC inhibitors alleviate the production of Aβ [37] and improve spatial memory in APP-overexpressing mice [38]. Notably, Class I and II HDAC inhibitors suppress morphological defects associated with zebrafish *Chd7* mutations [39], and Class I HDAC inhibitors suppress deficits in evoked neurotransmission, movement [40], and endocytosis [41] in *kis* mutants. *Kis* is the *Drosophila* ortholog of *CHD7* [42] and *CHD8*. Collectively, these data suggest that there may be a functional relationship between CHD proteins and APP. 

We sought to investigate whether CHD proteins and APP mediate similar synaptic processes in mature neurons using the *Drosophila* NMJ, which is a glutamatergic synapse similar to mammalian central nervous system synapses [43]. *Drosophila* is widely used to examine disease processes and synaptic pathology because of its non-redundant genome and conservation of molecular processes [44]. *Drosophila* expresses a single APP, APPL, which includes the conserved extracellular domains and the intracellular C-terminal domain but lacks the Aβ sequence [45]. Human APP can be cleaved by the fly γ-secretase [46,47] and APPL processing by secretases produces secreted fragments, an Aβ-like neurotoxic peptide, and an AICD [48]. We found that *kis* loss-of-function mutants and animals expressing human APP and BACE in their central nervous system share some, but not all, synaptic phenotypes examined. Although Kis does not regulate *appl* expression, Kis and APPL regulate the synaptic localization of Rab11 with mutations in both producing a similar loss of Rab11 as that of the single mutants. In contrast, simultaneous mutations in *kis* and *appl* produce additive deficits in larval locomotion suggesting that while Kis and APPL may cooperatively regulate some synaptic functions, they also possess unique synaptic functions.

## 2. Results

### 2.1. Expression of APP and BACE in Neurons Partly Phenocopies Loss of Function Mutations in Kis

CHD proteins affect the expression of genes and gene families identified as risk factors for neurodegenerative diseases [28,49]. We investigated the relationship between Kis and APPL by first examining synaptic characteristics regulated by Kis. Kis promotes the postsynaptic localization of glutamate receptors [50], the expression and synaptic localization of the GTPase Rab11, and endocytosis [41]. We examined these phenotypes in a fly model of AD, which was generated by expressing human APP [46] and its cleaving enzyme, β-secretase/β-site APP cleaving enzyme (BACE) [47], in the *Drosophila* central nervous system (CNS) using the *elav-Gal4* driver. Expression of these transgenes in the CNS leads to the deposition of amyloid plaques and degeneration of photoreceptor cells [47] and neurons in the mushroom bodies [51,52] of adult flies. Similar to *kis^LM27^/kis^k13416^* loss of function mutants [41,50], expression of APP and BACE in the CNS produced significant reductions compared to controls in synaptic levels of the glutamate receptor subunit, GluRIIC (Figure 1A), and in endocytosis as indicated by internalization of the lipophilic dye, FM 1-43FX (Figure 1B), at the third instar larval 6/7 NMJ. Similarly, synaptic levels of Rab11, the GTPase that regulates trafficking between recycling endosomes and the plasma membrane [53], was also significantly reduced in larvae expressing APP and BACE in the CNS (Figure 1C). 

Transforming growth factor (TGF)-β signaling is prevalent across metazoans and regulates diverse processes involved in embryogenesis, development, and adult tissue maintenance [54]. Expression of the downstream TGF-β component, SMAD4, is increased in AD patient brains and is a proposed biomarker for the disease [55]. Kis binds near or within the coding region of several components of the TGF-β/BMP signaling pathway including *mothers against dpp* (*mad*), *thick veins* (*tkv*), and *wishful thinking* (*wit*) in *Drosophila* intestinal stem cells [56]. Therefore, we examined levels of the phosphorylated R-SMAD, Mad (pMad), which is phosphorylated after TGF-β receptor binding to a ligand [43]. Synaptic levels of pMad were increased both in *kis^LM27^/kis^k13416^* mutants compared with controls (Figure 2A) and in animals expressing APP and BACE in the CNS compared with outcrossed controls (Figure 2B). Collectively, these data indicate that expression of APP and BACE in the CNS phenocopies *kis* loss of function mutants.

We next used two-electrode voltage clamp electrophysiology to more broadly assess the synaptic function of *kis* mutants and APP;BACE-expressing larvae. Neurotransmission was compromised in *kis* mutants as previously reported [41,50]. Similar to *kis^LM27^/kis^k13416^* mutants, larvae expressing APP and BACE in the CNS showed reduced amplitudes of spontaneous miniature endplate junctional currents (mEJCs) (Figure 3B,D), consistent with the loss of synaptic GluRIIC (Figure 1A). APP- and BACE-expressing animals also showed a decrease in mEJC frequency, indicating a reduction in the number of active zones, but this was unchanged in *kis* mutants (Figure 3D, right). The latter, however, exhibited decreased evoked EJC (eEJC) amplitudes, which were unchanged in APP;BACE-expressing larvae compared to *UAS-APP;BACE* outcrossed controls (Figure 3A,C). 

To determine whether the deficits in neurotransmission were sufficient to produce behavioral deficits, we examined the locomotor activity of *kis* mutants and APP;BACE-expressing larvae. Larval locomotor behavior is produced by CNS central pattern generators that send cholinergic inputs to presynaptic motor neurons [57]. The frequency and duration of presynaptic motor neuron activity positively correlate with postsynaptic muscle contractile force [58]. Both *kis^LM27^/kis^k13416^* mutants and larvae expressing APP and BACE in the CNS exhibited reductions in the distance traveled/average path length and maximum velocity of movement compared with controls (Figure 3E). Thus, APP;BACE-expressing larvae phenocopy some, but not all, characteristics of *kis* loss of function mutant synapses. In support of this, we noted that synaptic levels of the endocytic proteins Dynamin (Dyn) and Endophilin A (EndoA), which are enriched at *kis^LM27^/kis^k13416^* NMJs [41], showed no differences in APP;BACE-expressing larvae (Dyn: *elav-Gal4/+* = 1.00 ± 0.06 AU, n = 14; *UAS-APP;BACE/+* = 1.06 ± 0.03 AU, n = 14; *elav>APP;BACE* = 0.94 ± 0.07 AU, n = 14, *p* = 0.39. EndoA: *elav-Gal4/+* = 1.00 ± 0.06 AU, n = 14; *UAS-APP;BACE/+* = 1.05 ± 0.07 AU, n = 14; *elav>APP;BACE* = 1.10 ± 0.10 AU, n = 14, *p* = 0.66). These collective findings indicate that Kis, APP, and BACE likely have shared and non-shared roles at the synapse.

### 2.2. Kis Promotes the Synaptic Localization of APPL 

Based on our data, we hypothesized that Kis could promote synaptic function and the expression and localization of synaptic proteins including Rab11 by restricting the expression of *appl* (Figure 4A). Kis binds near the *appl* transcription start site in *Drosophila* intestinal stem cells [56] and *APP* is differentially expressed in *Chd8* heterozygous loss of function mutations in adult mice [59]. Alternatively, it is possible that AICD, generated by APP/APPL processing, promotes *kis* expression (Figure 4B). AICD is stabilized by the cytoplasmic adapter protein Fe65, transported to the nucleus, and associates with the histone acetylase Tip60 to form the AFT complex in HEK293 and neuroblastoma cells [60]. In addition, AICD and Kis may function in a complex (Figure 4C) to cooperatively regulate some synaptic functions by regulating the expression of synaptic gene products like *rab11*. Both AICD [61] and CHD7 regulate the expression of *TP53* [62], which encodes the transcription factor, p53. 

To begin to distinguish between these models, we examined APPL expression in *kis^LM27^/kis^k13416^* mutants. We found that *kis^LM27^/kis^k13416^* mutants exhibited a significant decrease in APPL protein at the NMJ (Figure 5A) but a 1.8-fold increase in *appl* transcripts in the CNS (Figure 5B). These data support a role for Kis in promoting APPL protein but not mRNA expression. Next, we examined synaptic function in *appl^d^* mutants, a null mutant [63], and *tip60^e02395^*, a hypomorph generated by the insertion of a *piggybac* element in the coding region of *tip60* [64]. If mutations in *appl* and *tip60* produce similar phenotypes as *kis* mutants, then Kis, APPL, and Tip60 may function in a complex to regulate the expression of genes important for synaptic function. Both *appl^d^* and *tip60^e02395^* mutants showed reductions in eEJC amplitudes (Figure 5C,E) and these were not significantly different from the reduction in eEJC amplitudes in *kis^LM27^/kis^k13416^* mutants. Similarly, *kis^LM27^/kis^k13416^* and *tip60^e02395^* mutants showed reduced mEJC amplitudes (Figure 5D,E), and *kis^LM27^/kis^k13416^* and *appl^d^* mutants showed a decrease in quantal content, which is an estimation of the number of vesicles released by the presynaptic motor neuron [65]. There was no change in mEJC frequency in any of the genotypes examined (*w^1118^* = 2.40 Hz, n = 12; *kis^LM27^/kis^k13416^* = 2.53 Hz, n = 12, *p* = 0.66; *appl^d^* = 3.00 Hz, n = 13, *p* = 0.24; *tip60^e02395^* = 2.43 Hz, n = 17, *p* = 0.93). Although these data support the hypothesis that Kis, APPL, and Tip60 cooperate to ensure proper neurotransmission, it is notable that *kis* mutants show a decrease in APPL protein at the synapse but an increase in *appl* mRNA. Thus, Kis may not regulate *appl* transcription but instead may influence the localization of APPL to the synapse. 

### 2.3. Disrupting APPL Processing Impairs Neurotransmission and Locomotion in Controls but Not Kis Mutants

If Kis regulates the synaptic localization of APPL, then interfering with APPL processing would not exacerbate the deficits in neurotransmission and locomotion observed in *kis* mutants. To examine this possibility, we used L-685,458, which binds to γ-secretase thereby proportionally decreasing Aβ production in adult rat hippocampal slices [66] and preventing the formation of the AFT complex in HEK293 cells [60]. Raising larvae on food containing 100 nM L-685,458 partially suppresses deficits in movement and significantly increases synaptic mitochondria and localization of the postsynaptic scaffold, Discs Large, to NMJs in third instar larvae expressing APP;BACE in the CNS [67]. Exposing larvae to L-685,458 throughout development resulted in decreased eEJC amplitudes for *w^1118^* controls but not for *kis^k13416^* or *kis^LM27^/kis^k13416^* mutants (Figure 6A,B). These data support previous work demonstrating that APP processing is important for evoked neurotransmission and synaptic plasticity [68,69]. There were no differences, however, in mEJC amplitudes, mEJC frequency, or quantal content of controls or *kis^LM27^/kis^k13416^* mutants raised in food containing the γ-secretase inhibitor, L-685,458.

To distinguish the developmental effects of inhibiting APPL processing from those in mature neurons, we exposed third-instar larvae to 100 nM L-685,485 for 24 h. Further, inhibiting γ-secretase for at least two days induces lysosomal dysfunction followed by endosomal dysfunction in mouse embryonic fibroblasts [70]. Exposing larvae to L-685,485 for 24 h was sufficient to produce locomotor deficits including reductions in average path length, average and maximum velocity, and body lengths per second in treated *w^1118^* larvae compared with those exposed to an equal volume of DMSO (Figure 6C). The same deficits were not observed in *kis^LM27^/kis^k13416^* mutants but were retained in *kis^k13416^* mutants. Thus, both evoked neurotransmission and inhibition of γ-secretase activity resulted in locomotor deficits in controls but did not affect these phenotypes in *kis^LM27^/kis^k13416^* mutants. Collectively, these data indicate that Kis acts upstream of APPL to promote neurotransmission and larval locomotor behaviors.

### 2.4. Kis and APPL Promote the Synaptic Localization of Rab11

Synaptic Rab11 is decreased in *kis* mutants [41] and in larvae expressing APP;BACE in the CNS (Figure 1C). Therefore, it is possible that both APPL and Kis regulate *rab11* expression. To test this possibility, we assessed synaptic Rab11 levels in *appl^d^* mutants and in animals heterozygous for mutations in both *appl* and *kis*. *Appl^d^* mutants exhibit a reduction in Rab11 at the synapse similar to that of *kis^LM27^/kis^k13416^* mutants (Figure 7A,B). If APPL and Kis function in a transcriptional complex to regulate synaptic Rab11 levels, we would expect that the loss of Rab11 in animals containing mutations in both *appl* and *kis* would be similar as that observed in animals with mutations in either *appl* or *kis*. Consistent with this, larvae with heterozygous loss of function mutations in both *appl* and *kis* show a decrease in synaptic Rab11 that is comparable to that of *appl^d^/+* and *kis^LM27^/+* mutants. In addition, knockdown of APPL in *kis^LM27^/kis^k13416^* mutants did not exacerbate the loss of Rab11 observed at *kis* mutant NMJs (Figure 7C,D).

To more broadly assess whether APPL and Kis work cooperatively to regulate synaptic function, we examined locomotor behavior in animals containing mutations in both *appl* and *kis*. Both *kis^LM27^/kis^k13416^* and *appl^d^* mutants demonstrated impaired locomotion with reductions in average path length and average and maximum velocities (Figure 7E). Larvae with mutations in both *kis^LM27^* and *appl^d^* mutants showed additive effects compared with both *appl^d^/+* and *kis^LM27^/+* mutants. Collectively, these data indicate that while Kis and APPL regulate common synaptic targets like Rab11, they also regulate non-overlapping synaptic targets that contribute to the generation and execution of locomotor behaviors.

Chromatin-remodeling enzymes influence the expression of thousands of genes [56,59,71]. Thus, the reduction in APPL at *kis^LM27^/kis^k13416^* mutant synapses (Figure 5A) likely results from the collective action of many differentially expressed gene products important for synaptic function. Therefore, to test the relationship between Rab11 and APPL, we manipulated Rab11 expression and activity in a control genetic background. If APPL promotes *rab11* transcription, we would expect that changing the expression or activity of Rab11 would not affect synaptic levels of APPL. If, however, Rab11 regulates the localization of APPL by determining the recycling of APPL, then changing the expression or activity of Rab11 would affect synaptic levels of APPL. In support of the latter, knockdown of Rab11 in presynaptic motor neurons using the *C380-Gal4* driver resulted in a reduction in synaptic APPL while knockdown of Rab11 in postsynaptic muscle using the *24B-Gal4* driver did not affect APPL (Figure 8A). Expression of wild-type Rab11 tagged with enhanced yellow fluorescent protein (Rab11^eYFP^) in either presynaptic motor neurons or postsynaptic muscle increased APPL at the NMJ. Similarly, expression of a constitutively active Rab11, which does not hydrolyze its bound GTP (Rab11^Q70L^), also increased synaptic APPL when expressed in presynaptic motor neurons or postsynaptic muscle (*UAS-Rab11^Q70L^/+* = 1.00 ± 0.07 AU, n = 15; *C380-Gal4/+* = 0.80 ± 0.07 AU, n = 14; *C380> Rab11^Q70L^* = 1.28 ± 0.19 AU, n = 13, *p* = 0.03; *24B-Gal4/+* = 0.99 ± 0.09 AU, n = 13; *24B> Rab11^Q70L^* = 2.08 ± 0.18 AU, n = 13, *p* < 0.0001). These data indicate that the reduction in synaptic APPL in *kis* mutants may occur due to reduced levels of Rab11. 

## 3. Discussion

Chromatin-regulatory proteins establish and maintain the epigenome, the collective set of chemical modifications that occur on DNA and histones. The epigenome integrates both cell-intrinsic programs and environmental cues [72] to regulate gene expression by determining chromatin structure. The CHD proteins, CHD7 and CHD8, are best known for their roles in neurodifferentiation and neurodevelopment [27,28] but are expressed in mature neurons [29,30]. Epigenetic changes in gene expression underlie persistent synaptic remodeling that occurs in learning and memory [73] and cognitive decline [74]. The transcriptional profiles from individuals with autism spectrum disorders or AD illustrate shared and distinct sets of affected neuronal genes [75] suggesting that these processes share a subset of altered molecular pathways. 

### 3.1. Kis and APPL Functionally Interact and Localize Rab11 to the Synapse

We investigated the link between chromatin remodeling and AD by examining synaptic phenotypes in *Drosophila* larvae expressing human APP and BACE in their CNS and comparing them with *kis* mutants. Loss of function *kis* mutants and animals expressing human APP and BACE in their CNS exhibit reduced synaptic levels of the glutamate receptor subunit, GluRIIC, the GTPase Rab11 (Figure 1A,C), and the BMP component, pMad (Figure 2), at the *Drosophila* larval NMJ. Collectively, these synaptic abnormalities, along with others not investigated here, likely produced reductions in endocytosis (Figure 1B), neurotransmission, and movement of *kis^LM27^/kis^k13416^* mutants and APP;BACE-expressing larvae (Figure 3). Some characteristics of neurotransmission and the localization of the endocytic proteins EndoA and Dyn, however, differed between *kis* mutants and APP;BACE-expressing larvae. Collectively, these data suggest that there may be a functional interaction between Kis and APPL to influence some aspects of synaptic function.

It is possible that the functional interaction between Kis and APPL occurs because Kis restricts *appl* transcription in the larval CNS (Figure 4A). Several transcriptional programs including those downstream of growth factors [76], androgens [77], cytokines [78], apolipoprotein E, and extracellular regulated kinase (ERK) [79] promote *APP* transcription in neurons. Transcription of a subset of ERK targets is enhanced by interactions between CHD8 and the transcription factor, ELK1, in mature human neurons [80]. In *Drosophila* intestinal stem cells, Kis restricts expression of a negative regulator of the epidermal growth factor receptor, which is upstream of ERK, and binds near the *appl* transcription start site [56]. Alternatively, Kis may be a target of the AFT complex (Figure 4B), formed by APPL’s AICD, which interacts with the adapter protein FE65 in the cytoplasm, translocates to the nucleus, and binds to the histone acetylase TIP60 [81]. Although *appl* mRNA was increased in the CNS of *kis^LM27^/kis^k13416^* mutants (Figure 5B), there was a decrease in APPL at the NMJ synapse (Figure 5A). These data do not support a model where Kis restricts *appl* transcription in the larval CNS. Instead, Kis may indirectly influence APPL localization via direct transcriptional regulation of other synaptic proteins as Kis differentially regulates hundreds of synaptic targets [50]. Kis also promotes trafficking through the endomembrane system [82], a process that is disrupted in AD and other neurodegenerative diseases [83]. The loss of APPL at *kis* mutant synapses may, therefore, result from aberrant intracellular trafficking and/or a compensatory response that downregulates *appl* translation [84].

It is also possible that Kis is recruited to the AFT complex (Figure 4C) to regulate the transcription of genes that encode synaptic proteins thereby promoting synaptic function. Chromatin remodeling proteins form unique multisubunit complexes by associating with different combinations of transcription factors and chromatin modifying enzymes in a cell- and developmental-dependent manner [85,86]. Neurotransmission at *appl^d^* and *tip60^e02395^* mutant synapses shared some features with *kis^LM27^/kis^k13416^* mutants (Figure 5C–E). Further, *kis^LM27^/kis^k13416^* mutants do not show exacerbated deficits in evoked neurotransmission and locomotion after inhibition of APPL processing using L-685,458 [60]. Inhibition of APPL processing in controls, however, decreased evoked neurotransmission and locomotion (Figure 6). These data support a role for Kis in AFT complex-mediated transcription of synaptic targets that enable proper locomotion and neurotransmission. These data are also consistent, however, with a model where Kis indirectly influences APPL localization via direct transcriptional regulation of other synaptic and/or trafficking proteins. 

*Rab11* may be a target of a potential Kis–AFT complex as Kis and APPL promoted the synaptic localization of Rab11 and simultaneous loss of function mutations in each did not produce an additive loss of synaptic Rab11 (Figure 7A). Similarly, the knockdown of APPL in the *kis^LM27^/kis^k13416^* mutant background did not produce a further decrease in Rab11 (Figure 7B). Although these data are consistent with a Kis–AFT complex regulating *rab11* transcription, it is also important to consider the possibility that Kis regulates synaptic levels of APPL indirectly by promoting *rab11* transcription [87]. Rab11 is associated with intracellular vesicles containing APP in mouse forebrain neurons [88] suggesting that Rab11, at least partly, determines the concentration of APPL localized to the plasma membrane. Rab11 also directly interacts with the BACE subunits Presenilin 1 and Presenilin 2 [89], transports BACE1 along hippocampal dendrites and axons [90], and promotes the production of Aβ [91]. Thus, Kis may influence the synaptic localization of APPL by promoting *rab11* transcription thereby increasing synaptic levels of Rab11 and, as a result, APPL. In support of this, tknockdown of Rab11 in presynaptic motor neurons reduced synaptic APPL (Figure 8A) while overexpression of Rab11 (Figure 8B) or expression of a constitutively active Rab11 in either presynaptic motor neurons or postsynaptic muscle increased synaptic APPL. 

### 3.2. APPL and Rab11 May Influence the Synaptic Localization of One Another

Our results also illustrate a potentially reciprocal relationship between Rab11 and APPL. *Appl^d^* mutants exhibited decreased synaptic levels of Rab11 (Figure 7A,B) and Rab11 influenced the localization of APPL. Knockdown of Rab11 decreased, while overexpression of Rab11 increased, synaptic levels of APPL (Figure 8). While Rab11 traffics APPL to and from the plasma membrane [88], the localization of Rab11 may be partly determined by APPL processing. Processing of APP generates Aβ, which hydrolyzes the phosphatidylinositol, PI(4,5)P_2_ by activating type I metabotropic glutamate receptors and phospholipase C [92]. Although Rab11 is recruited to membranes enriched in PI(3,5)P_2_ [93], PI(3,5)P_2_ and PI(4,5)P_2_ share the precursor, PI4P [94]. Therefore, the membrane concentrations of PI(3,5)P_2_ and PI(4,5)P_2_ are partly dependent on one another. Thus, APPL processing may restrict synaptic levels of Rab11. 

Collectively, our results support a model where Kis acts upstream of APPL to regulate some aspects of synaptic function. Our conclusions, however, are limited because we cannot exclude the possibility that Kis is recruited to the AFT complex to cooperatively regulate the transcription of synaptic gene products. Indeed, our data are consistent with a combination of the two models where Kis and APPL work cooperatively to promote *rab11* expression and Kis acts upstream, through Rab11, to properly localize APPL to the synapse. We are further limited because the phenotypes we observe in *kis* mutants and animals expressing APP and BACE in their CNS are the sum of hundreds to thousands of differentially expressed gene products. Kis influences the transcription of thousands of potential targets [56] and the expression of hundreds of genes is affected in AD [20]. Despite this, it is important to use models that emulate the complexity of the disease process to gain a better understanding of AD.

### 3.3. Aberrant Chromatin Remodeling Contributes to AD

Chromatin remodeling contributes to neurodifferentiation and neurodevelopment [95], learning and memory [73], healthy aging [96], and neurodegeneration [25]. Our data add to a growing body of work indicating that aberrant chromatin remodeling is a driving force in AD. Genome-wide changes in histone acetylation influence transcriptional regulation and likely contribute to the differential expression of genes in AD. Genome-wide hypo- and hyperacetylation of H3K27 is observed in AD patient entorhinal cortex genomes compared with non-diseased age-matched controls [17] and increased H3 acetylation precedes cognitive impairments in an AD mouse model. Similarly, AD patients show a net loss of H4K16 acetylation in the temporal lobe [97]. Kis promotes multiple histone modifications including H3K27 [98] and H4K16 acetylation [87]. Thus, the similarities we observed between *kis* mutants and APP- and BACE-expressing larvae may occur due to the loss of histone acetylation.

Histone acetylation and other chromatin modifications are dynamic processes [23] and AD pathology is characterized by the dysregulation of hundreds of gene products [17,18,19,20]. Thus, targeting gene expression globally by altering neuronal epigenomes offers another potential therapeutic strategy for treating AD patients. Therapeutic approaches like targeting Aβ and inhibiting γ-secretase activity have shown limited success. The Food and Drug Administration (FDA) recently approved the use of lecanemab and donanemab, which are monoclonal Aβ antibodies, for the treatment of early AD. These compounds slightly delayed cognitive decline in patients with early AD but ultimately do not slow the production of Aβ [99] or restore the synapses or neurons that are lost when patients seek treatment [15]. γ-secretase inhibitors, which include L-685,458, limit the production of Aβ [66] but produce side effects as chronic administration inhibits Notch signaling [100]. Chromatin modifications also occur in cancer epigenomes and several different HDAC inhibitors are FDA approved to treat hematological malignancies. With the continued development of and several HDAC inhibitors in clinical trials [101], it is promising to investigate the of potential of repurposing these compounds. However, we need to better understand the specific changes in chromatin structure that occur in AD, when they occur in the disease process, and how they contribute to disease pathology.

## 4. Materials and Methods

### 4.1. Drosophila Stocks and Husbandry

Flies were raised at 25 °C in a Percival DR-36NL incubator with a 12 h light:dark cycle in standard vials containing Jazz Mix (Fisher Scientific AS153; Hampton, NH, USA) food. The following fly stocks were obtained from the Bloomington *Drosophila* Stock Center: *24B-Gal4* (BL 1767), *appl^d^* (BL 43632), *C380-Gal4* (BL 80580), *elav-Gal4* on X (BL 458), *elav-Gal4* on chromosome II (BL 8765), *kis^k13416^* (BL 10442), *tip60^e02395^* (BL 18052), *UAS-appl^RNAi^* (BL 28043), *UAS-Rab11^Q70L^* (BL 50783), *UAS-Rab11^eYFP^* (BL 50782), *UAS-Rab11^RNAi^* (BL 27730), and *w^1118^* (BL 3605), which was used as the isogenic control for most experiments. Both *kis^LM27^* [102] and *UAS-APP;BACE* [67] were obtained from the laboratory of Dr. Dan Marenda. *Kis^LM27^/kis^k13416^* was generated by crossing the respective stocks because *kis^LM27^* is an embryonic lethal mutation. Outcrossed controls for experiments using *UAS* transgenes and *Gal4* drivers were generated by crossing each respective stock with *w^1118^*. Both male and female flies and larvae were used for all experiments except for those including *appl^d^* and *tip60^e02395^*. *Appl* and *tip60* are located on the X chromosome, necessitating the use of females for these experiments.

Larvae treated with 100 nM L685,458 (Tocris 2627; Bristol, United Kingdom) were raised on instant food (Genessee Scientific Nutri-Fly 66–117; Morrisville, NC, USA) containing 1.6% of 10% *w*/*v* Tegosept (Genessee Scientific 20-258) for 24 h or for their lifetime. Controls were raised on food containing the same components except an equal volume of DMSO was added in place of L685,458. Food was made in small batches, stored at 4 °C, and used within five days of preparation. 

### 4.2. Immunocytochemistry and FM 1-43FX Labeling

Third-instar larvae were dissected at room temperature in Roger’s Ringer (135 mM NaCl, 5 mM KCl, 4 mM MgCl_2_*6H_2_O, 1.8 mM CaCl_2_*2H_2_O, 5 mM TES, 72 mM Sucrose, 2 mM glutamate, pH 7.15) on Sylgard (World Precision Instruments 501986; Sarasota, FL, USA)-lined petri dishes. Larvae were fixed in either Bouin’s Fixative (Ricca Chemical 112016; Arlington, TX, USA) or 3.7% formaldehyde freshly diluted from a 10× stock (37% Formaldehyde, Fisher Scientific BP531-500). After fixing for 30 min, larvae were placed in 1.5 mL centrifuge tubes containing PTX (1 × PBS + 0.1% Triton X-100, Fisher Scientific AAA16046AP) and washed three times for 10 min. Larvae were then washed two times for 30 min in PBTX (1 × PBS + 0.1% Triton X-100 + 1% Bovine Serum Albumin, Fisher Scientific BP1600-100) and incubated at 4 °C overnight in PBTX containing primary antibodies. Primary antibodies included guinea pig α-APPL (1:250, generated for this study), mouse α-Dynamin (1:300, BD Biosciences 610245; Franklin Lakes, NJ, USA), guinea pig α-Endophilin A (1:5000, Hugo Bellen lab), rabbit α-GluRIIC (1:5000, generated by Genscript using the sequence described in [103]), rabbit α-pMad (1:100, Abcam ab52903; Cambridge, United Kingdom), and mouse α-Rab11 (1:50, BD Biosciences 610657). The following day larvae were washed three times for 10 min and two times for 30 min in PBTX. Then, secondary antibodies including α-mouse FITC (113-095-003), α-mouse TRITC (115-025-003), α-rabbit FITC (106-095-003), or α-guinea pig FITC (111-095-003) were applied at 1:400 along with either Cy3-HRP (123-165-021) or A647-HRP (123-605-021) at 1:125 in PBTX for 2 h. All secondary antibodies and HRP were obtained from Jackson Immunoresearch Laboratories. Larvae were washed three times for 10 min and two times for 30 min in PBTX, placed on slides, and covered with VectaShield mounting medium (Vector Laboratories H1000; Newark, CA, USA).

Polyclonal APPL antibodies were generated by Genscript by optimizing and expressing the full-length *appl* cDNA sequence in *E. coli*. Purified APPL protein was injected into guinea pigs with boosters administered at two, four, and six weeks after initial injection. Seven weeks after primary immunization, serum was collected from animals. The serum was affinity purified and the APPL antibody was tested by comparing synapse-specific immunolabeling with total guinea pig IgG in both control (*w^1118^*) and *appl^d^* null larvae.

FM 1-43FX (Fisher Scientific F35355) labeling was performed as previously described [104]. Third-instar larvae were dissected in Ca^2+^-free HL-3 (100 mM NaCl, 5 mM KCl, 10 mM NaHCO_3_, 5 mM HEPES, 30 mM sucrose, 5 mM trehalose, 10 mM MgCl_2_, pH 7.2) in Sylgard-coated petri dishes. Once larvae were fillet dissected, they were washed once with Ca^2+^-free HL-3, the motor neurons were cut, and 4 μM FM 1–43FX in HL-3 containing 90 mM KCl and 1.5 mM CaCl^2^ was applied for 1 min. Residual/unbound FM 1-43FX was washed off by replacing the Ca^2+^-free HL-3 five times over 5–10 min. Larvae were then fixed with 3.7% formaldehyde in Ca^2+^-free HL-3 for 5 min and subsequently transferred to 1.5 mL centrifuge tubes. Larvae were washed 10 times over 10–15 min with Ca^2+^-free HL-3 containing 2.5% goat serum and then incubated at room temperature in A647-HRP in Ca^2+^-free HL-3 containing 2.5% goat serum for 30 min. Finally, larvae were washed with Ca^2+^-free HL-3 10 times over 10–15 min, placed on slides, covered with Vectashield, and immediately imaged via confocal microscopy. 

Larval 6/7 NMJs of segments A3 or A4 were imaged using the 60× oil immersion objective of an Olympus Fluoview 1000 microscope. Image acquisition parameters were determined based on the mean fluorescence intensities used for control larvae. Approximately equal numbers of control and experimental animals were imaged on the same day. Max projected z-stacks were used for image analyses, which were performed using Fiji [105]. Mean pixel intensities were obtained from immunolabeled images by outlining the NMJ and subtracting the background pixel intensity obtained from an area of equal size that did not include the NMJ from the value that represented the NMJ. For FM 1-43FX labeling, relative fluorescence was obtained for each z-stack slice by subtracting background fluorescence from NMJ fluorescence. The mean of all z-stacks was used and reported for FM 1-43FX intensity.

### 4.3. RNA Isolation and RT-qPCR

Third-instar larvae CNSs were dissected at room temperature in Roger’s Ringer (135 mM NaCl, 5 mM KCl, 4 mM MgCl_2_*6H_2_O, 1.8 mM CaCl_2_*2H_2_O, 5 mM TES, 72 mM Sucrose, 2 mM glutamate, pH 7.15) on Sylgard (World Precision Instruments 501986)-filled 60 mm petri dishes. Isolated CNSs were placed in nuclease-free 1.5 mL centrifuge tubes containing RNAlater solution (Fisher Scientific AM7020) and stored at −20 °C until RNA was isolated from tissues using an Invitrogen RNAqueous Total RNA Isolation Kit (Fisher Scientific AM1912). RNA concentrations were assessed using an Implen NanoPhotometer N50. RT-qPCR was performed using the iTaq Universal SYBR Green One-Step Kit (BioRad 1725151) and a CFX Connect Real-Time PCR Detection System (Bio-Rad; Hercules, CA, USA). 100 ng of RNA and 50 pmol/µL of cDNA-specific primers were added to each reaction. Each technical replicate included 30 CNSs and two biological replicates were used to calculate relative fold changes. Relative fold changes were calculated by determining the difference between the C(t) value of the target transcript reaction and the C(t) value for GAPDH to obtain ΔC(t) for each transcript. Next, the difference between control and *kis* mutant ΔC(t)s was calculated and log transformed to obtain the 2^−ΔΔ^C(t) [106].

### 4.4. Electrophysiology and Behavior

Third-instar larvae were fillet dissected in ice-cold HL-3 + 0.25% CaCl_2_ by gluing larval pelts to Sylgard-coated 18 mm round coverslips. The CNS was removed by cutting the motor neurons and the HL-3 was replaced with room temperature HL-3 + 1.0% CaCl_2_. Recordings were acquired from muscle 6 of segment A3 or A4 using glass microelectrodes with 10–20 MΩ of resistance filled with 3 M KCl. Muscles with input resistances < 5 MΩ were voltage-clamped at −60 mV by an Axoclamp 900A amplifier (Molecular Devices; San Jose, CA, USA). Segmental nerves were stimulated by a fire-polished glass electrode filled with room temperature HL-3 + 1.0% CaCl_2_ and a Grass S88 stimulator with a SIU5 isolation unit (A-M Systems; Sequim, WA, USA). Recordings were digitized with a Digidata 1443 digitizer (Molecular Devices) and data were analyzed in Clampfit (v 11.1, Molecular Devices). Three minutes of mEJCs were analyzed to determine mEJC amplitudes and frequencies. The mean amplitude of 10 eEJCs elicited by suprathreshold stimuli was used to represent eEJC amplitudes. Quantal content was calculated by dividing the integrated area of eEJCs by that of mEJCs [107]. Approximately equal numbers of controls and experimental animals were used daily for recordings.

Larvae used for locomotor assays were transferred from vials to a 100 mm plate containing 1.6% agar. They were allowed to acclimate to the crawling surface and shed home vial debris for 1 min. Next, larvae were placed on a 1.6% agar arena and locomotion was recorded after animals began moving. Locomotor behavior was recorded for five animals simultaneously on a Canon EOS M50 camera at 29.97 frames per second. Video recordings were analyzed in Fiji using the wrMTrck plugin (written by Jesper S. Pedersen). 

### 4.5. Experimental Design and Statistical Analyses

Two to four biological replicates including 3–8 animals were used for each experiment. Approximately equal numbers of controls and experimental animals were used for each biological replicate. The total number of technical replicates is represented by the points (circles, squares, triangles, or diamonds) found associated with histogram bars. Data analyses were performed with GraphPad Prism (v. 10.2.3). Unpaired *t*-tests were used to analyze data from experiments that included a single control group while one-way ANOVAs followed by Tukey’s or Bonferroni’s post hoc tests were used to analyze data from experiments that included more than one control group. Bartlett’s test for homogeneity of variance was used to assess the variances between data sets. Statistical significance is represented on bar graphs as follows: * = <0.05, ** = <0.01, *** = <0.001 with error bars representing standard error of the mean (SEM).

## Figures and Tables

**Figure 1 ijms-25-08429-f001:**
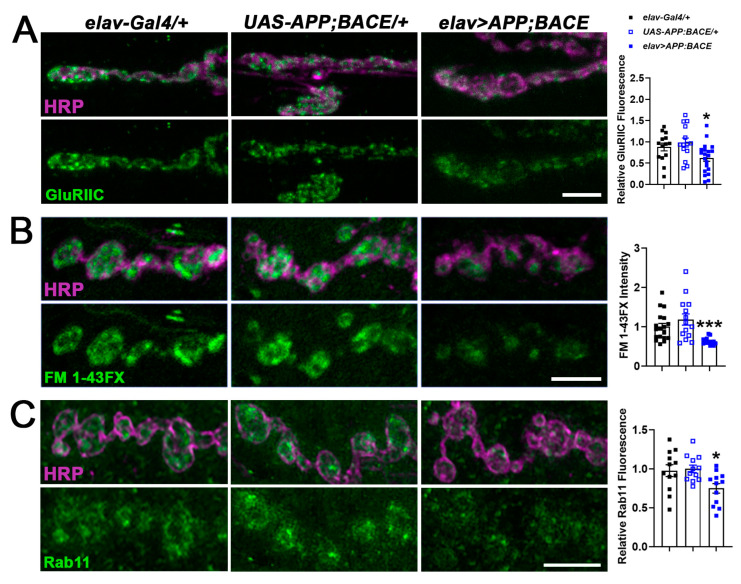
Expression of APP and BACE in the larval CNS mimics *kis* loss of function mutant phenotypes. High-resolution confocal micrographs showing the glutamate receptor subunit, GluRIIC (**A**, green), endocytosis as indicated by FM 1-43FX (**B**, green), and the GTPase Rab11 (**C**, green) at the 6/7 NMJ in the listed genotypes. Fluorescence intensities, relative to controls, are depicted in right histograms. HRP (magenta) denotes the presynaptic motor neuron. Scale bars = 5 µm. * = *p* < 0.05, *** = *p* < 0.001.

**Figure 2 ijms-25-08429-f002:**
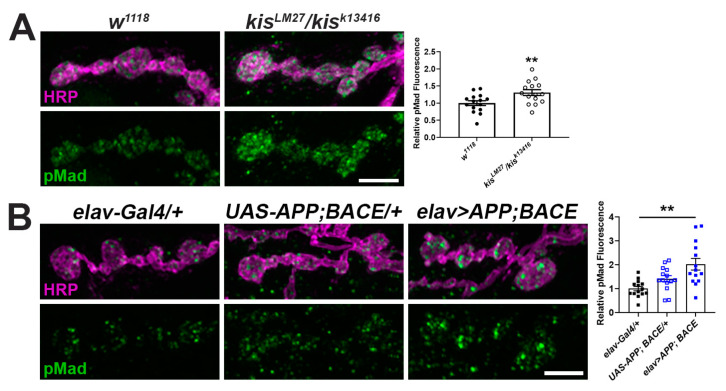
There is an increase in the TGF-β/BMP signaling component pMad at *kis* mutant and APP and BACE expressing NMJs. (**A**,**B**) High-resolution confocal micrographs of the 6/7 NMJ of genotypes listed showing synaptic pMad (green) and HRP (magenta), which labels the presynaptic motor neuron. Fluorescence intensities, relative to controls, are depicted in right histograms. Scale bars = 5 µm. ** = *p* < 0.01, F = 9.24.

**Figure 3 ijms-25-08429-f003:**
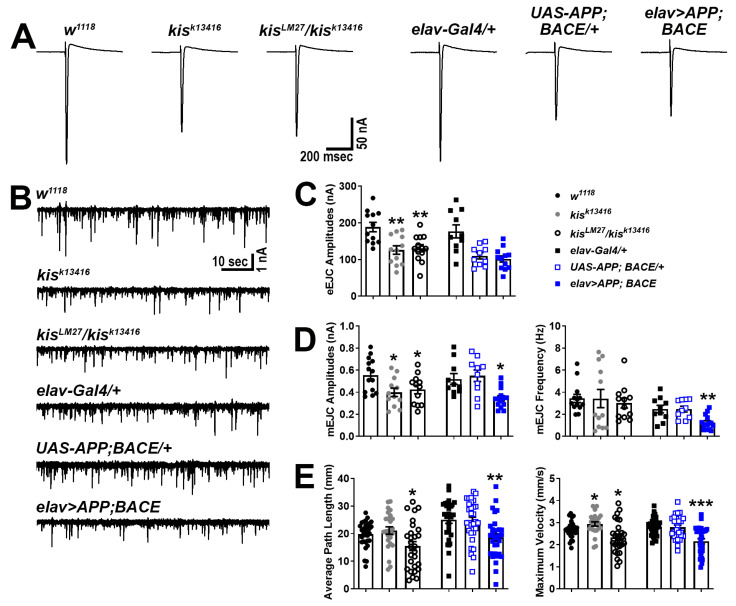
*Kis* mutants and APP;BACE-expressing larvae show deficits in neurotransmission and locomotion. (**A**) Representative eEJC traces recorded from muscle 6 in response to administration of a suprathreshold stimulus to the presynaptic motor neuron. (**B**) Representative mEJC traces from larval body wall muscle 6. (**C**) Quantification of eEJC amplitudes. ** = *p* < 0.01 (**D**) Quantification of mEJC amplitudes (left) and frequency (right). * = *p* < 0.05, F = 8.50; ** = *p* < 0.01, F = 9.24 (**E**) wrMTrck quantification of larval crawling behavior on an agar arena for 30 s. Distance traveled or average path length (left) and maximum crawling velocity (right) were normalized to body size (body lengths per second). * = *p* < 0.05; ** = *p* < 0.01, F = 6.89; *** = *p* < 0.001, F = 15.14.

**Figure 4 ijms-25-08429-f004:**
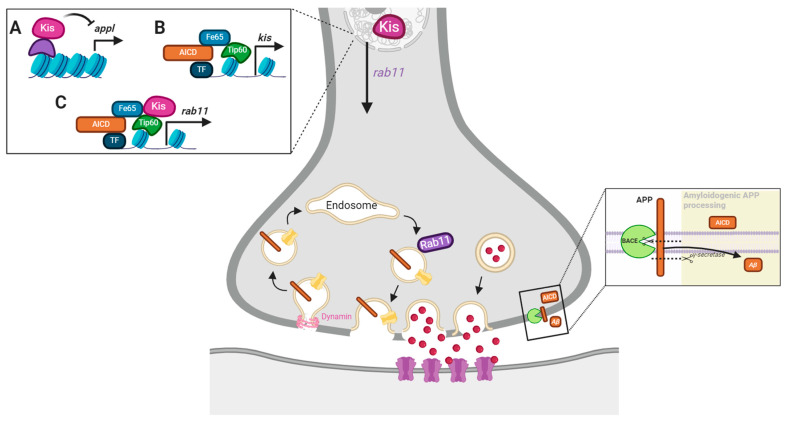
Model of potential Kis and APPL interactions. Both Kis and AICD are transcriptional regulators. Therefore, Kis may influence synaptic function by repressing neuronal APPL (**A**), being a target of AICD-mediated transcription (**B**), or being recruited by the AFT complex to chromatin to promote *rab11* transcription (**C**).

**Figure 5 ijms-25-08429-f005:**
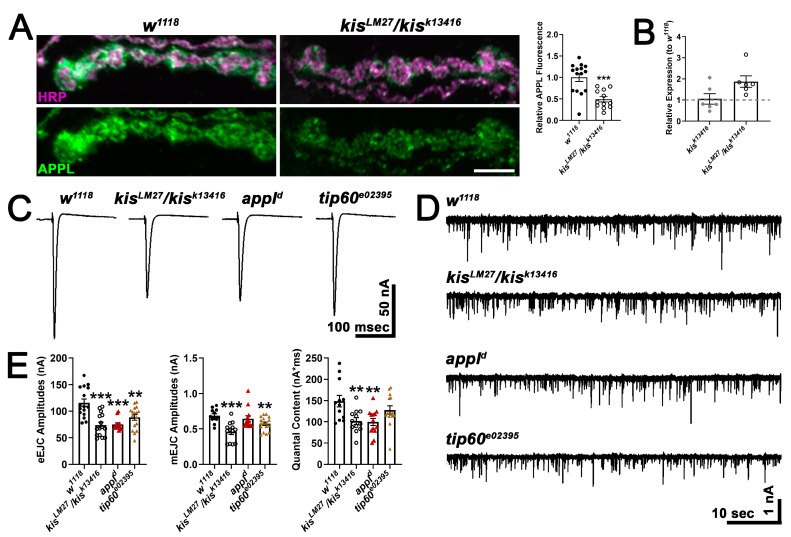
Kis regulates APPL localization and may act in a transcriptional complex with AICD and Tip60. (**A**) High-resolution confocal micrographs of terminal 6/7 boutons immunolabeled for APPL (green). HRP (magenta) was used to delineate presynaptic motor neurons. Scale bar = 5 µm. Right histogram shows the quantification of relative APPL fluorescence intensity. *** = *p* < 0.001 (**B**) Histogram showing *appl* transcript levels in *kis* mutants relative to *w^1118^* controls. (**C**) Representative eEJC traces recorded from muscle 6 of third instar larvae in response to administration of a suprathreshold stimulus to the presynaptic motor neuron. (**D**) Representative mEJC traces from larval body wall muscle 6. (**E**) Quantification of eEJC amplitudes (left), mEJC amplitudes (middle), and quantal content (right). ** = *p* < 0.01, *** = *p* < 0.001.

**Figure 6 ijms-25-08429-f006:**
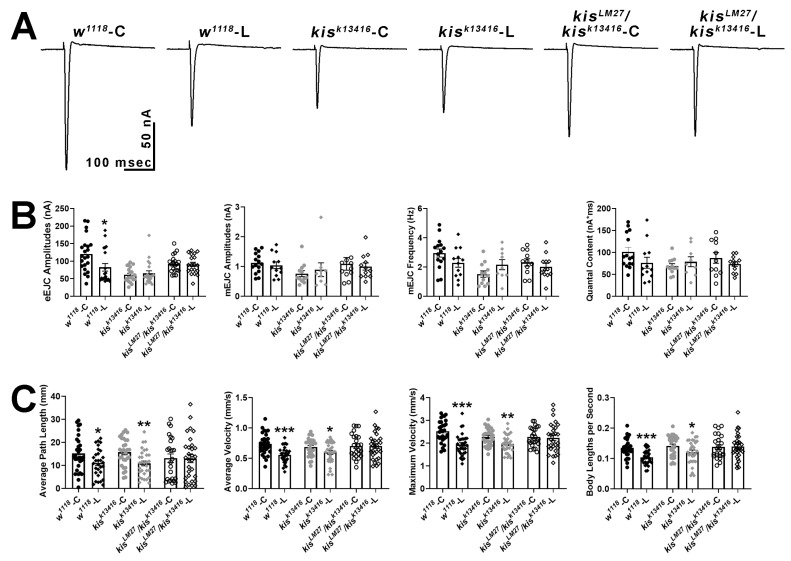
Pharmacological inhibition of γ-secretase activity with L-685,458 impairs evoked neurotransmission and locomotion in controls but not in *kis^LM27^/kis^k13416^* mutants. (**A**) Representative eEJC traces recorded from muscle 6 of third instar larvae in response to administration of a suprathreshold stimulus to the presynaptic motor neuron. C represents animals exposed to the vehicle, DMSO, while L represents animals exposed to 100 nM L-685,458 throughout development. (**B**) Quantification of eEJC amplitudes, mEJC amplitudes and frequency, and quantal content. * = *p* < 0.05 (**C**) wrMTrck quantification of larval crawling behavior on an agar arena for 30 s. C represents animals exposed to the vehicle, DMSO, while L represents animals exposed to 100 nM L-685,458 for 24 h. Histograms show average path length, average and maximum velocities, and body lengths per second. * = *p* < 0.05, ** = *p* < 0.01, *** = *p* < 0.001.

**Figure 7 ijms-25-08429-f007:**
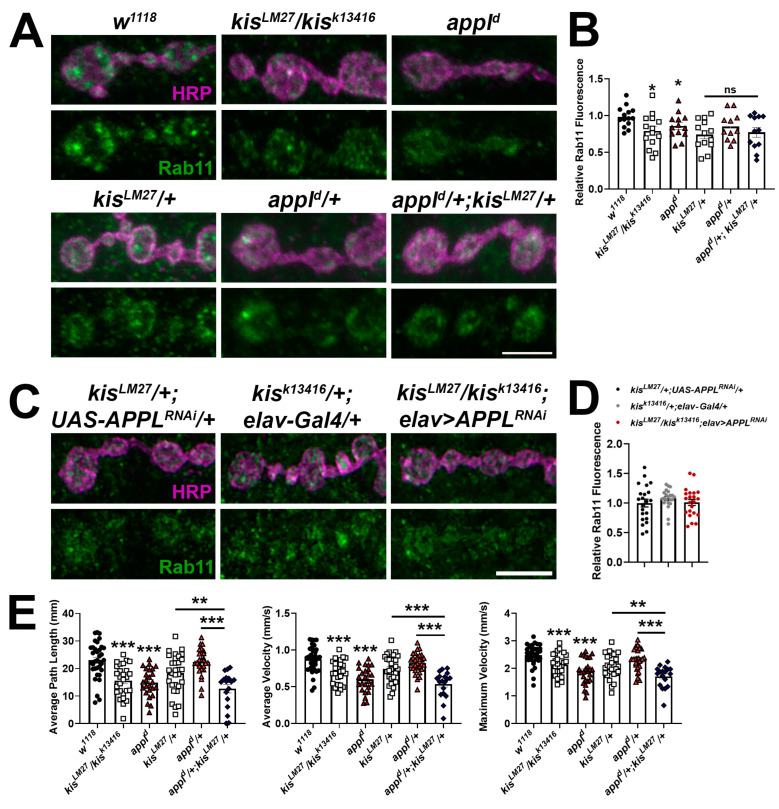
APPL and Kis may cooperatively regulate the synaptic localization of Rab11. (**A**) High-resolution confocal micrographs of terminal 6/7 boutons immunolabeled for Rab11 (green). HRP (magenta) was used to label presynaptic motor neurons. Scale bar = 5 µm. (**B**) Histogram showing relative Rab11 fluorescence intensities. * = *p* < 0.05 (**C**) High-resolution confocal micrographs of terminal 6/7 boutons immunolabeled for Rab11 (green). HRP (magenta) was used to denote presynaptic motor neurons. Scale bar = 5 µm. (**D**) Histogram showing relative Rab11 fluorescence intensities. (**E**) wrMTrck quantification of larval crawling behavior on an agar arena for 30 s. Histograms show average path length and average and maximum velocities normalized to body lengths per second. * = *p* < 0.05, ** = *p* < 0.01, *** = *p* < 0.001.

**Figure 8 ijms-25-08429-f008:**
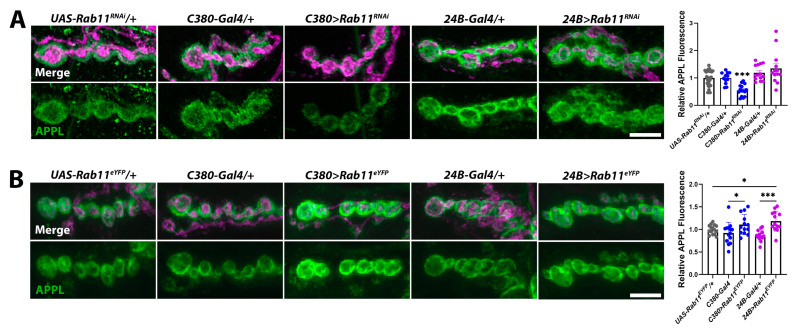
Rab11 regulates synaptic levels of APPL. (**A**) High-resolution confocal micrographs of terminal 6/7 boutons immunolabeled for APPL (green). HRP (magenta) was used to label presynaptic motor neurons. (**A**) Rab11 was knocked down in presynaptic motor neurons using the *C380-Gal4* driver and in postsynaptic muscle using the *24B-Gal4* driver. Scale bar = 5 µm. Right histogram shows APPL fluorescence intensities. *** = *p* < 0.001, F = 6.51 (**B**) Rab11 was overexpressed in presynaptic motor neurons by expressing *UAS-Rab11^eYFP^* using the *C380-Gal4* driver and in postsynaptic muscle using the *24B-Gal4* driver. Scale bar = 5 µm. Right histogram showing relative APPL fluorescence intensities. * = *p* < 0.05, F = 4.61; *** = *p* < 0.001, F = 14.05.

## Data Availability

The data presented in this study are available upon request from the corresponding author. Data are available in the form of spreadsheets documenting data analysis of raw data.

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
