# Peer review of "Kismet/CHD7/CHD8 and Amyloid Precursor Protein-like Regulate Synaptic Levels of Rab11 at the Drosophila Neuromuscular Junction"

_ijms, 2024, doi:10.3390/ijms25158429_

Round 1

Reviewer 1 Report

Comments and Suggestions for Authors

In the current study, the authors present experimental results determining the synaptic function of chromodomain helicase DNA proteins and amyloid precursor proteins. For the study, the authors used the Drosophila neuromuscular junction model. The latter, due to its similarity to the mammalian glutamatergic synapse, can be considered an excellent model of changes occurring during Alzheimer disease. The paper is excellently prepared and edited, and in principle I have no major comments.

Minor aspects:

Line 88 - some abbreviations such as HDAC appear only once in the text. Therefore, I do not see the point of introducing them.

Line 106 - "bone morphogenetic protein" instead of "bone morphogenic protein".

Line 455 - for the equipment and reagents used, it is advantageous to specify their country of origin, possibly codes. Authors must remember that other researchers should be able to reproduce their experiment.

Line 518 - the main shortcoming of the work is the question of checking the specificities of the primary antibodies used. How was this tested? There is a lack of positive, negative or presdorption control tests.

Author Response

Line 88 - some abbreviations such as HDAC appear only once in the text. Therefore, I do not see the point of introducing them.

We agree with the reviewer. HDAC appears three times in lines 88-90 and subsequent times in the discussion but we removed the abbreviation for Discs Large. The latter was the only abbreviation was only used once in the text.

Line 106 - "bone morphogenetic protein" instead of "bone morphogenic protein".

We thank the reviewer for catching this and have made the correction.

Line 455 - for the equipment and reagents used, it is advantageous to specify their country of origin, possibly codes. Authors must remember that other researchers should be able to reproduce their experiment.

We included the source for all reagents by providing the company name and catalog numbers in the methods section. This information enables other researchers to purchase the same reagents we used in our experiments. All equipment and reagents were purchased from companies in the United States. Given that a researcher can identify the reagents we used by searching for the company name and catalog number, we felt that listing “USA” for every company was redundant and unnecessary.

Line 518 - the main shortcoming of the work is the question of checking the specificities of the primary antibodies used. How was this tested? There is a lack of positive, negative or presdorption control tests.

All antibodies used in this study were previously validated and used in previous publications. Applicable references were provided in the text. The exception to this was the APPL antibody. We described the generation and testing of this antibody in the methods including, “The serum was affinity purified and the APPL antibody was tested by comparing syn-apse-specific immunolabeling with total guinea pig IgG in both control (w1118) and appld null larvae.” (lines 502-504).

Reviewer 2 Report

Comments and Suggestions for Authors

This original research work reports on the transmembrane protein β-amyloid precursor protein (APP) which is central to the pathophysiology of Alzheimer’s disease. The hypothesis is that aberrant processing of APP leads to the formation of neurotoxic β-amyloid aggregates, that consequently impairs cognitive function in AD. The authors have focused on a better understanding of the synaptic function of APP. They have acknowledged the limitations of this study and the inconclusive results based on the findings.

There are a couple of points that the authors are encouraged to address in the revised version.

1) related to ethical board review: the authors have mentioned: “Not applicable”. Does this mean that the authors did not obtain the approved protocol with the use of models in this study? Why?

2) the pharmacological tools have been used here by the authors to understand the mechanisms. However, the authors can elaborate further on whether and how some of these pharmacological tools can be used later in translational view for use as therapeutic agents or not.

3) Please add the novelty of this study and how in light of already existing literature can add information that progresses the field further. It is also important that the authors also add the implication of these findings.

Author Response

1) related to ethical board review: the authors have mentioned: “Not applicable”. Does this mean that the authors did not obtain the approved protocol with the use of models in this study? Why?

Drosophila is an invertebrate model organism. Use of invertebrate model organisms does not require approval by an Institutional Animal Care and Use Committee (IACUC) or by the National Institutes of Health, which funded the study.

2) the pharmacological tools have been used here by the authors to understand the mechanisms. However, the authors can elaborate further on whether and how some of these pharmacological tools can be used later in translational view for use as therapeutic agents or not.

We used a single pharmacological tool, L-685,458, to determine whether Kis acted upstream of APPL. We have added a paragraph to the discussion that describes the therapeutic use of γ-secretase inhibitors like L-685,458. These compounds are no longer used clinically because of their side effects, which were largely attributed to chronic administration inhibiting Notch signaling. γ-secretases recognize over 140 potential substrates (Hur, J.Y. (2022). γ-Secretase in Alzheimer’s disease. Experimental and Molecular Medicine, 54: 433-446.). We sought to limit the effects of γ-secretase inhibition on other molecular pathways by acute exposure to L-685,458 as described in the results (lines 295-298).

3) Please add the novelty of this study and how in light of already existing literature can add information that progresses the field further. It is also important that the authors also add the implication of these findings.

We have added two paragraphs to the discussion to highlight the growing body of literature that illustrates the importance of epigenetic modifications to AD disease pathology. Our work compliments several other studies and reinforces the possibility of using epigenetic modifying compounds to treat AD.

Reviewer 3 Report

Comments and Suggestions for Authors

The manuscript of Hendricks et al. investigates the potential role of APP in the neuromuscular junction. The Drosophila APP-like (APPL) has both shared and non-shared roles at the synapse with Kismet (Kis), a chromatin helicase binding domain (CHD) protein. Kis likely influences the synaptic localization of APPL, possibly by promoting rab11  transcription. In the introduction, the role of APP in AD is presented. In the results section, the authors explore synaptic characteristics and the relation between Kis and APPL and use clamp electrophysiology to elucidate the role of the Kis. The discussion highlights the strength of the model presented by the authors. 

Comments:

Line 57-58 Further, focusing on APP and Aβ production has demonstrated little 57 therapeutic benefit for AD patients [16]- please clarify and elaborate on it

Lines 104-116 would fit better into disussion

The introduction is quite long and chaotic, an illustration could help to facilitate the understanding of the text. Maybe the introduction may be shortened as well

Figure 1: the quality could be better, i there any possibility of replacing this figure with one of the better quality?

Please describe the weaknessess of the study

It would be helpful to comment on the Drosophila as a model of AD

Author Response

Line 57-58 Further, focusing on APP and Aβ production has demonstrated little 57 therapeutic benefit for AD patients [16]- please clarify and elaborate on it

In light of the reviewer’s subsequent comments about the length of the introduction, we didn’t elaborate on this information in the introduction. Instead, we briefly describe the limited success of these therapeutics in the final paragraph that we added to the discussion.

Lines 104-116 would fit better into discussion

The reviewer is referring to the final paragraph of the introduction, which typically includes a summary of the results. We condensed this summary based on the reviewer’s next recommendation.

The introduction is quite long and chaotic, an illustration could help to facilitate the understanding of the text. Maybe the introduction may be shortened as well

We thank the reviewer for providing this feedback and have condensed the introduction. Although the introduction is still >750 words, we felt this was necessary given the likely audience for this work. AD researchers may be more familiar with the clinical aspects of the disease necessitating a description of the synaptic mechanisms of the disease. AD researchers and neurobiologists may have limited knowledge of epigenetics necessitating a brief description of chromatin remodeling. We further needed to link chromatin remodeling to AD and justify the use of Drosophila as an experimental model for a human disease.

Figure 1: the quality could be better, i there any possibility of replacing this figure with one of the better quality?

We are not sure what the reviewer means by “quality” given that the figure represents fluorescence microscopy, which is also shown in Figures 2, 5, 7, and 8. Because the reviewer did not suggest the latter figures needed to be improved, we are inferring that “quality” refers to the specificity/diffuse nature of the Rab11 immunosignal and/or the grainy appearance of the FM 1-43FX signal. The weak, grainy appearance of FM 1-43FX was previously noted and attributed to reduced fluorescence compared to other FM dyes as a result of fixing the molecule [1]. We selected images for Figure 1 and all figures that were representative of mean relative intensities. In addition, our images are comparable to other published images using FM 1-43FX [1] and Rab11 [2].

Please describe the weaknessess of the study

Weaknesses/limitations of the of the study are described in the last paragraph of section 3.2 of the discussion.

It would be helpful to comment on the Drosophila as a model of AD

Drosophila as a model of AD is described in the last paragraph of the introduction and first paragraph of the results.

  1. Verstreken, P., Ohyama, T. & Bellen, H. J. FM 1-43 labeling of synaptic vesicle pools at the Drosophila neuromuscular junction. Methods Mol Biol 440, 2008. 349-369, doi:10.1007/978-1-59745-178-9_26.
  2. Latcheva, N.K., et al., The CHD Protein, Kismet, is Important for the Recycling of Synaptic Vesicles during Endocytosis. Sci Rep, 2019. 9(1): p. 19368.

Reviewer 4 Report

Comments and Suggestions for Authors

The manuscript “Kismet/CHD7/CHD8 and Amyloid Precursor Protein Like Regulate Synaptic Levels of Rab11 at the Drosophila Neuromuscular Junction” by Linskey is a research article which examined the role of the transmembrane protein β-amyloid precursor protein (APP) in synaptic function. The authors found that Drosophila APP-like (APPL) has both shared and non-shared roles at the synapse with Kismet (Kis), a chromatin helicase binding domain (CHD) protein. The authors also demonstrated that loss of function mutations in kis and animals expressing human APP and BACE in their central nervous system show reductions in the glutamate receptor subunit, GluRIIC, the GTPase Rab11, and the bone morphogenic protein (BMP), pMad at the Drosophila larval neuromuscular junction (NMJ). In these animals, processes like endocytosis, larval locomotion, and neurotransmission are deficient. These findings indicate that there is a functional relationship between Kis and APPL but Kis does not regulate appl expression at the larval NMJ whereas Kis seems to influence the synaptic localization of APPL, possibly by promoting rab11 transcription. In general, this research article is critical in this field and contains essential findings. However, I have several comments before this manuscript is accepted for publication.

1. In my opinion, it would be better to place Figure 4 as the last figure. At the last figure, summary findings could be displayed.

2. In this study, ANOVA was used for statistical assessment. Please add the F values in text or figure legends.

3. line 575: “Unpaired t-texts” to “Unpaired t-tests?”.

Author Response

  1. In my opinion, it would be better to place Figure 4 as the last figure. At the last figure, summary findings could be displayed.

We also thought about including Figure 4 as the last figure. However, we refer to three potential models of Kis and APPL interactions after Figure 3. Therefore, we felt it would be best to show those models as illustrations/Figure 4 instead of simply describing the potential interactions.

  1. In this study, ANOVA was used for statistical assessment. Please add the F values in text or figure legends.

The F-values for all ANOVAs were added to figure legends.

  1. line 575: “Unpaired t-texts” to “Unpaired t-tests?”.

We thank the reviewer for catching this typo and have made the necessary edit.

Round 2

Reviewer 3 Report

Comments and Suggestions for Authors

Thank you for the revisions. I had a second look at the Figure 1 and it is fine.

I recommend the manuscript for publication.